# Plasma Versus Whole Blood Tacrolimus Concentrations and Health-Related Quality of Life in Kidney Transplant Recipients

**DOI:** 10.3390/pharmaceutics17050590

**Published:** 2025-04-30

**Authors:** Svea Nolte, J. Casper Swarte, Tim J. Knobbe, Ilja M. Nolte, Tanja R. Zijp, Harmen R. Moes, Marco van Londen, Niels L. Riemersma, Johannes R. Björk, Rinse K. Weersma, Gea Drost, Stefan P. Berger, Daan J. Touw, Stephan J. L. Bakker

**Affiliations:** 1Division of Nephrology, Department of Internal Medicine, University Medical Center Groningen, University of Groningen, 9713 GZ Groningen, The Netherlands; s.nolte@umcg.nl (S.N.); j.c.swarte@umcg.nl (J.C.S.); t.j.knobbe@umcg.nl (T.J.K.); m.van.londen@umcg.nl (M.v.L.); n.l.riemersma@umcg.nl (N.L.R.); s.p.berger@umcg.nl (S.P.B.); 2Department of Neurology, University Medical Center Groningen, University of Groningen, 9713 GZ Groningen, The Netherlands; h.r.moes@umcg.nl (H.R.M.); g.drost@umcg.nl (G.D.); 3Department of Gastroenterology and Hepatology, University Medical Center Groningen, University of Groningen, 9713 GZ Groningen, The Netherlands; r.j.bjork@umcg.nl (J.R.B.); r.k.weersma@umcg.nl (R.K.W.); 4Department of Epidemiology, University Medical Center Groningen, University of Groningen, 9713 GZ Groningen, The Netherlands; i.m.nolte@umcg.nl; 5Department of Clinical Pharmacy and Pharmacology, University Medical Center Groningen, University of Groningen, 9713 GZ Groningen, The Netherlands; t.r.zijp@umcg.nl (T.R.Z.); d.j.touw@umcg.nl (D.J.T.); 6Department of Neurosurgery, University Medical Center Groningen, University of Groningen, 9713 GZ Groningen, The Netherlands; 7Department of Pharmaceutical Analysis, Groningen Research Institute of Pharmacy, University of Groningen, 9712 GZ Groningen, The Netherlands

**Keywords:** tacrolimus, kidney transplant recipients, health-related quality of life, therapeutic drug monitoring

## Abstract

**Background/Objectives**: Tacrolimus dosing traditionally relies on therapeutic drug monitoring in whole blood, while assessment in plasma may better reflect its effect and reveal overdosing impacting health-related quality of life (HRQoL). **Methods**: In this cross-sectional study, 898 kidney transplant recipients (KTRs) who were at least 12 months post-transplantation were included. Plasma and whole blood tacrolimus concentrations were compared using Passing–Bablok regression analyses and Bland–Altman plots. Furthermore, the relationship with daily tacrolimus dose and with HRQoL (mental component summary (MCS), physical component summary (PCS)) was explored using linear regression by comparing standardized coefficients. Lastly, mediation analyses explored the effect of various tacrolimus-related side effects on the association between tacrolimus concentrations and HRQoL. **Results**: Comparison of the methods revealed a constant bias and a slight proportional bias between whole blood and plasma tacrolimus concentrations. The Bland–Altman plots indicated poor agreement with a statistically significant ratio difference (*p* < 0.001). Both whole blood and plasma concentrations were significantly associated with daily tacrolimus dose (both *p* < 0.001). Compared to whole blood tacrolimus concentrations, plasma tacrolimus concentration showed a strong negative association with worse HRQoL (PCS: st. β = −0.12, *p* = 0.01; MCS: st. β = −0.14, *p* < 0.001). The associations between plasma tacrolimus concentrations and HRQoL were mediated by fatigue severity (proportion mediated on PCS: 67.8%, MCS: 59.5%) and reduced kidney function (proportion mediated on PCS: 16.7%, MCS: 12.9%). **Conclusions**: In conclusion, compared with whole blood tacrolimus concentrations, plasma tacrolimus concentrations exhibited a negative association with HRQoL in KTRs. Consequently, therapeutic drug monitoring using plasma tacrolimus concentrations may reduce the occurrence of tacrolimus-related toxicity.

## 1. Introduction

Tacrolimus has been the cornerstone of immunosuppression after kidney transplantation since the 1990s and has played a leading role in improving the 5-year survival rates in kidney transplant recipients (KTRs) [1]. At the same time, tacrolimus is known for its wide range of adverse events, in particular, nephrotoxicity, neurotoxicity, and the development of post-transplantation diabetes mellitus (PTDM) and gastrointestinal toxicity [2]. As a result, the use of tacrolimus can have a tremendous impact on patients’ morbidity and health-related quality of life (HRQoL) [3].

Tacrolimus binds extensively to erythrocytes and circulating plasma proteins including albumin and α_1_-acid glycoprotein, leaving only 0.3–2% as the free active fraction in plasma [4]. The narrow therapeutic window and large variety of side effects, together with the high inter- and intra-patient pharmacokinetic variability of tacrolimus, require therapeutic drug monitoring (TDM) [2]. Due to limitations of the analytical techniques available at the time of introduction of tacrolimus as an immunosuppressive drug, assessment of plasma concentrations was not yet feasible. Thus, TDM has historically relied on the assessment of whole blood concentrations [5]. During the last decade, the relationship between whole blood tacrolimus concentrations and outcomes has been studied extensively, along with refinement of the therapeutic range for whole blood tacrolimus TDM to minimize adverse effects and improve effectiveness [5]. However, the use of whole blood trough levels for TDM is currently debated since the relationship with clinical outcomes is not always apparent [5,6]. From a clinical perspective, determining either the free effective fraction of tacrolimus or the concentration within its active compartment, i.e., within T-cells, seems to be preferable [7,8]. However, determining the free fraction of tacrolimus or intracellular concentrations remains methodologically challenging and is not easily standardized [9,10,11,12]. Due to advances in analytical methodology, a possible solution is now to determine plasma tacrolimus levels, which linearly correlate with unbound tacrolimus plasma concentrations [4].

Therefore, the current study investigated whether plasma tacrolimus levels exhibited a stronger association with HRQoL than whole blood tacrolimus levels in KTRs. First, we compared plasma and whole blood tacrolimus levels and daily tacrolimus doses. Second, we compared the relationship between plasma and whole blood tacrolimus levels and HRQoL and explored the mediating effect of various tacrolimus-related side effects.

## 2. Materials and Methods

### 2.1. Patient Population and Study Design

The present study falls within the scope of the TransplantLines Biobank and Cohort Study at the University Medical Center Groningen (UMCG), the Netherlands (ClinicalTrials.gov identifier: NCT03272841) [13]. Participants include new transplant candidates, previously transplanted patients who received their graft prior to the start of TransplantLines, and (potential) living kidney or liver donors. All participants provided written informed consent prior to their inclusion. The study was approved by the local Medical Ethical Committee (METc 2014/077). All procedures were in accordance with the Declaration of Helsinki and the Declaration of Istanbul. The manuscript was prepared following the Strengthening the Reporting of Observation Studies in Epidemiology (STROBE) guidelines [14]. For the current study, we included data of KTRs with a functional graft for at least 12 months post-transplant and documented tacrolimus use. Subjects were excluded if blood samples were hemolytic or if peak levels of tacrolimus were determined.

For included subjects, demographic and clinical data were extracted from medical records. Clinical examinations were performed during a study visit at the outpatient clinic. On the day of inclusion, blood samples were obtained after an overnight fasting period of 8 to 12 h and before morning dosage of tacrolimus.

Most transplant recipients received tapering triple-immunosuppressive therapy consisting of prednisolone, mycophenolate mofetil, and tacrolimus. Tacrolimus dosing was individualized through TDM to achieve whole blood trough concentrations of 4–6 ng/mL from six months or longer after transplantation.

### 2.2. Tacrolimus Analyses

Prior to the study visit, 10 mL EDTA blood samples were collected for both whole blood tacrolimus monitoring and tacrolimus plasma analyses. Beforehand, patients were instructed to administer tacrolimus in case of a once-daily dosing regimen at 10:00 a.m. and for a twice-daily dosing regimen at 10:00 a.m. and 10:00 p.m. Blood sampling was generally performed at ≈8:00 a.m., which was considered to be ≈10 h or ≈22 h after the last drug administration. Whole blood samples were analyzed within 4 h and were processed to plasma within the same time frame by centrifugation at 1300× *g* for 10 min at room temperature (20 °C). Plasma samples were subsequently stored at −80 °C until analysis, which was performed within 3 freeze–thaw cycles. The lower limit of quantification (LLOQ) of the method was 0.05 µg/L, while for some patients the tacrolimus plasma levels were below this threshold, but tacrolimus could still be detected. To avoid exclusion of these patients and since measurements below the LLOQ are unreliable, the imputation method of 1/2 of the LLOQ was chosen. Thus, plasma concentrations were set at 0.025 µg/L [15]. All tacrolimus analyses were performed at the Laboratory of Clinical Pharmacy and Pharmacology at the UMCG using a TSQ Quantiva Mass Spectrometer with Vanquish UHPLC system, both from Thermo Fisher Scientific (Waltham, MA, USA), using a validated high-throughput LC-MS/MS method [16,17,18]. Zijp et al. published the precision data for the used method [18].

### 2.3. Health-Related Quality of Life

To assess HRQoL, participants completed the 36-item Short-Form Health Survey (SF-36) [19]. The SF-36 can be divided into eight subdomains: physical functioning, role limitations due to physical health problems, bodily pain, general health perceptions, energy/fatigue or vitality, social functioning, role limitations due to personal or emotional problems, and general mental health [20]. All scales range from 0 to 100, with higher scores indicating higher levels of functioning or well-being.

These subdomains can be aggregated into two summary scores: the physical component summary (PCS) and the mental component summary (MCS). The component summary scores were calculated using a standardized three-step procedure: all eight subscale scores were standardized using a linear z-score transformation using normative data from the 1998 US general population [21]. Next, the z-scores were multiplied by the subscale factor score coefficients derived from the oblique factor rotation, which allows for correlation of the physical and mental health constructs, and were subsequently summed. Finally, t-scores were calculated by multiplying the obtained PCS and MCS sums by 10 and adding 50 to the product, to yield a mean of 50 and a standard deviation of 10 for the US norm population [22].

### 2.4. Tacrolimus-Related Side Effects

The following section outlines the data collection methods and procedures used to assess the various tacrolimus-related side effects. To assess fatigue, participants completed the Checklist Individual Strength 20-revised (CIS20-R), which was developed as a self-reported multidimensional instrument to assess qualitatively distinct and relevant aspects of fatigue [23]. For the analyses in the current study, the subscale fatigue severity was used, which ranges from 8 to 56, with higher scores indicating a higher level of fatigue. Sleep quality was assessed using the Pittsburgh Sleep Quality Index (PSQI), a validated 19-item questionnaire that assesses seven components of sleep quality [24]. Duration of sleep, sleep disturbances, sleep latency, daytime dysfunction due to sleepiness, sleep efficiency, overall sleep quality, and sleep medication use were scored and combined into a final score ranging from 0 to 21, with higher scores indicating poorer sleep quality. Recently, it has been shown that KTRs suffer from dysbiosis, a disruption in the balance of the gut microbiome, and that immunosuppressive drugs are a driving factor [25]. Therefore, the Shannon Diversity Index, a measure of the overall diversity of the gut microbiome, which is often decreased in dysbiosis, was included in the current study as a tacrolimus-related patient outcome [26]. The Shannon Diversity Index was calculated on species abundance data obtained from previously performed fecal metagenomics. A detailed method section can be found in Swarte et al. [25]. Another side effect of tacrolimus is diarrhea, which is highly underreported in regular questionnaires in transplant patients [27]. Diarrhea was assessed by measuring stool water content, of which detailed methods are described in Douwes et al. [27]. As a representor for kidney function, the estimated glomerular filtration rate (eGFR) was calculated using the CKD-EPI formula, which is validated and differentiated by sex and plasma creatinine levels [28]. Creatinine analyses were performed using a Roche Cobas 8000 platform (Roche, Mannheim, Germany). Creatinine measurements were calibrated to the isotope-dilution mass spectrometry reference method. PTDM was defined as diabetes that developed in the post-transplant period, excluding transient post-transplant hyperglycemia that may occur in the early post-transplant period [29,30,31]. Since all participants in the current study were enrolled at least 12 months after transplantation, diabetes was defined as a fasting plasma glucose higher than or equal to 7.0 mmol/L, HbA1c above or equal to 48 mmol/mol, or the use of antidiabetic medication [29]. Analyses of fasting blood glucose were performed using a Roche Cobas 8000 platform (Roche, Mannheim, Germany), and HbA1c levels were determined using a Tosoh G8 (ion exchange HPLC) (Tosoh/Sysmex Europe, Norderstedt, Germany). Information on alopecia was obtained from the updated and validated 59-item Modified Transplant Symptom Occurrence and Symptom Distress Scale (MTSOSD-59R), in which subjects rated, amongst other symptoms, the occurrence of alopecia [32]. Tremor was assessed using a Dutch translation of the Fahn–Tolosa–Marin Tremor Rating Scale part C (TRS-C), which includes eight questions assessing patient-perceived tremor occurrence during activities of daily living [33]. Scores are summed to a maximum of 32, with higher scores indicating more severe tremor symptoms. The Fahn–Tolosa–Marin scale has been shown to be a valid tool for assessing tremor in clinical practice; for logistical reasons, only part C was evaluated in patients in the current study [34].

### 2.5. Statistical Analyses

Continuous variables with a normal distribution were described as mean ± standard deviation or as median [interquartile range] if the distribution was skewed. Histograms and Q-Q plots were used to assess the distribution of continuous variables. Categorical variables were presented as frequencies with percentages.

To compare the methods of tacrolimus whole blood concentrations with tacrolimus plasma concentrations and to investigate any linear relationship between the methods, Passing–Bablok regression analyses were performed. The Passing–Bablok regression method is a robust, non-parametric method, which is not sensitive to non-normally distributed errors and outliers in the data, making it a suitable approach for comparing measurements from different analytical methods allowing to determine any systematic bias between them [35]. Furthermore, to evaluate the agreement between whole blood tacrolimus concentrations and plasma tacrolimus concentrations, Bland–Altman absolute difference plots and Bland–Altman ratio difference plots were produced (the ratio was calculated as plasma tacrolimus/whole blood tacrolimus). For these analyses, scaled values of plasma and whole blood tacrolimus concentrations were calculated by dividing the respective concentration by the standard deviation of plasma or whole blood tacrolimus of KTRs. Assumptions of normality and homoscedasticity of the residuals were checked with visual evaluation of the Q-Q plot and residual plots, respectively [36]. We followed the guidelines from the European Medicines Agency (EMA), which state that any two methods can be considered to be in agreement if their difference is within ±20% around the mean difference for >67% of the samples [37].

To assess the relation between whole blood and plasma tacrolimus concentrations and the respective daily tacrolimus doses, linear regression analyses were performed. To prevent collinearity, in two separate regression models either whole blood or plasma tacrolimus concentration was tested as an independent variable. The residuals of the linear regression models were plotted and checked for normal distribution. Subsequently, to assess whether plasma tacrolimus concentration better reflect the daily tacrolimus dose than whole blood tacrolimus concentration, we tested whether the standardized regression coefficient of the former was significantly larger than that of the latter using bootstrapping with 1000 replications. The proportion of replications for which the standardized regression coefficient of plasma tacrolimus concentration was smaller than or equal to the standardized regression coefficient of whole blood tacrolimus concentration, determined the significance of this test.

To investigate the association of whole blood and plasma tacrolimus concentrations with the primary outcome PCS and MCS of HRQoL, the same methodology as described above was followed. We tested for effect modification by age and sex. Additionally, sensitivity analyses with adjustment of whole blood tacrolimus levels for hematocrit were performed since it has been shown that patients with low hematocrit exhibit a larger difference between whole blood and unbound tacrolimus concentrations [4]. The hematocrit-corrected whole blood tacrolimus model was compared with the crude model with plasma tacrolimus concentration. Furthermore, sensitivity analyses were added that assessed the intra-patient variability (IPV) of plasma tacrolimus concentrations and its association with the PCS and MCS of HRQoL. IPV was calculated by dividing the standard deviation by the mean of two repeated measurements of plasma tacrolimus concentrations per patient and multiplying by 100. This provided a coefficient of variability (COV) per patient. No repeated measurements of HRQoL were available for this cohort.

To unravel possible mediators for the association found between plasma tacrolimus and HRQoL, mediation analyses were performed with the following tacrolimus-related side effects as possible mediators: fatigue, sleep quality, gut microbiome dysbiosis, diarrhea, kidney function, PTDM, alopecia, and tremor. First, assumptions for mediation analyses were checked, which included assessing the significant associations between the potential mediator and plasma tacrolimus concentration and the association between the potential mediator and HRQoL controlling for plasma tacrolimus concentration. Mediation analyses were then performed according to the method of Preacher and Hayes [38,39]. The mediated proportion was calculated as the ratio of the indirect effect and total effect of plasma tacrolimus on HRQoL. It indicates the extent to which the mediating variable explains the total effect.

SPSS version 28 for Windows (IBM, Armonk, NY, USA) and R Statistical Software version 3.2.3 (R Foundation for Statistical Computing, Vienna, Austria) were used for statistical analyses. *p* values ≤ 0.05 were considered statistically significant. Due to logistical constraints inherent in conducting a large cohort study, some data points were missing for certain KTRs (as outlined in Section 3.1). For each analysis, only participants with complete data relevant to that specific analysis were included, resulting in varying sample sizes reported for each analysis.

## 3. Results

### 3.1. Study Population

In the current study, 898 KTRs were included, with a mean age of 55.1 ± 13.8 years (Table 1). One subject was excluded due to hemolysis of the blood sample, and 25 subjects were excluded because of measurement of tacrolimus peak levels (Figure 1).

### 3.2. Comparison of Tacrolimus Concentrations in Whole Blood and Plasma

Among KTRs, different tacrolimus formulations were prescribed: 573 KTRs (63.8%) used Prograf, 224 KTRs (24.9%) used Advagraf, 89 KTRs (9.9%) used Envarsus, and for 12 KTRs (1.3%), the formulation was unknown. The mean whole blood tacrolimus concentration was 5.7 ± 1.9 µg/L (range: 1.0–16.3) and mean plasma tacrolimus concentration was 0.13 ± 0.07 µg/L (range: 0.03–0.65). In total, 35 KTRs (4.3%) had a plasma tacrolimus concentration below 0.05 μg/L. The median plasma tacrolimus concentration of KTRs with BLOQ levels was 0.04 [0.03–0.05] μg/L (the actual measurement was unavailable for 1 KTR of the 35 KTRs with values BLOQ). When we repeated all the statistical analyses, with these BLOQ values included rather than the values set at 0.025 μg/L, all the results remained materially unchanged.

We compared tacrolimus measured in whole blood and plasma using Passing–Bablok regression analyses and Bland–Altman plots (Table 2 and Figure 2). The scaled whole blood tacrolimus versus plasma tacrolimus concentration showed a constant bias and a slight proportional bias. Bland–Altman analyses showed no absolute difference between scaled whole blood tacrolimus and scaled plasma tacrolimus concentrations, but a statistically significant ratio difference was observed (*p* < 0.001, Table 2 and Figure 3). In total, 36% of the paired concentrations were within ±20% of the mean ratio, not meeting the predefined minimum of >67% [37]. These results indicate poor agreement and therefore suggest that plasma and whole blood tacrolimus concentrations cannot be interpreted interchangeably.

### 3.3. Tacrolimus Concentrations and Dose Associations

The associations of whole blood and plasma tacrolimus concentrations with the tacrolimus dose were statistically significant (both *p* < 0.001), see Figure 4. When comparing the plasma tacrolimus model with the whole blood tacrolimus model, no statistically significant difference was found between the bootstrap standardized regression coefficients (Table 3).

### 3.4. Assessment of Tacrolimus Concentrations and Health-Related Quality of Life

To test the hypothesis that plasma tacrolimus concentrations exhibit a stronger negative association with HRQoL than whole blood concentrations, the standardized regression coefficients of the whole blood and plasma tacrolimus concentrations were compared using a bootstrap procedure.

As shown in Table 4, higher plasma tacrolimus concentrations were statistically significantly associated with lower PCS and MCS (*p* = 0.003, *p* < 0.001, respectively); however, no statistically significant association was found between whole blood tacrolimus concentrations and PCS and MCS. When comparing the plasma tacrolimus model with the whole blood tacrolimus model, the bootstrapped standardized coefficients for the associations of plasma tacrolimus with the PCS and MCS were significantly lower than those for whole blood tacrolimus (PCS: st. β = −0.12 vs. −0.03, *p* = 0.01; MCS; st. β = −0.14 vs. 0.001, *p* < 0.001). No effect modification by age or sex was found (see Appendix A, Table A1).

In sensitivity analyses, whole blood tacrolimus adjusted for hematocrit was not statistically significantly associated with PCS or MCS. When comparing the crude plasma tacrolimus model with the adjusted whole blood tacrolimus model, the results of the primary analysis were confirmed (see Appendix A, Table A2).

In additional sensitivity analyses, IPV was assessed in a subset of the study cohort. In 162 KTRs, tacrolimus plasma concentrations were measured at 12 months and 24 months after transplantation. There was no significant difference in the plasma concentration of tacrolimus at 12 and 24 months after transplantation (Wilcoxon signed-rank test, V = 8734, *p* = 0.20; see Appendix B, Figure A1A). In total, 71 KTRs had a COV > 30%, which is considered a high IPV (see Appendix B, Figure A1B) [40]. Next, we analyzed whether COV was associated with HRQoL by correlating COV with PCS and MCS. COV was not significantly associated with the PCS or MCS (r = 0.01, *p* = 0.88 and r = 0.04, *p* = 0.59, respectively, see Appendix B, Figure A1C,D).

### 3.5. Mediation Effects on Health-Related Quality of Life

To identify potential mediators of the statistically significant association found between plasma tacrolimus and the PCS and MCS of HRQoL, mediation analyses were performed.

Assessing the relationship between plasma tacrolimus concentration and the potential mediators revealed a statistically significant association with fatigue severity (st. β = 0.11, *p* = 0.01), gut microbiome dysbiosis (st. β = −0.14, *p* = 0.002), and kidney function (st. β = −0.14, *p* < 0.001). No association was found between plasma tacrolimus concentration and sleep quality, diarrhea, PTDM, alopecia, or tremor.

As shown in Figure 5, fatigue severity was found to be a partial mediator of the effect of plasma tacrolimus on the PCS (proportion mediated: 67.8%) and the MCS (proportion mediated: 59.5%), without significant direct effect. Furthermore, lower kidney function was found to be a partial mediator between plasma tacrolimus and the PCS (proportion mediated: 16.7%) and the MCS (proportion mediated: 12.9%). Importantly, the direct effect between kidney function and the PCS (st. β = −0.11; *p* = 0.03) and the MCS (st. β = −0.12, *p* = 0.004) remained statistically significant. Sleep quality, gut microbiome dysbiosis, diarrhea, PTDM, alopecia, and tremor did not function as mediators between plasma tacrolimus and the PCS and MCS of HRQoL. The detailed results can be found in Appendix C.

## 4. Discussion

In the current study, plasma tacrolimus levels were associated with worse HRQoL, whereas whole blood tacrolimus levels did not show a statistically significant association. We found that fatigue and reduced kidney function mediated part of the associations between plasma tacrolimus concentrations and HRQoL. Additionally, we demonstrated that plasma and whole blood tacrolimus levels cannot be used interchangeably due to bias and poor agreement between these two methods in KTRs. The daily administered tacrolimus dose was strongly associated with tacrolimus concentrations, measured in either whole blood or plasma.

When comparing the method of determining plasma tacrolimus levels to whole blood tacrolimus levels, Passing–Bablok regression and Bland–Altman ratio difference plots showed that plasma and whole blood tacrolimus cannot be used interchangeably. We demonstrated that high whole blood tacrolimus levels do not necessarily reflect high plasma tacrolimus levels. This conclusion is supported by the findings of Sikma et al., who showed a non-linear relationship between whole blood and unbound tacrolimus plasma concentrations in recipients early after thoracic organ transplantation, especially for higher whole blood concentrations [4]. Interestingly, both whole blood and plasma tacrolimus concentrations were significantly associated with daily tacrolimus dose; however, the association between particularly higher whole blood tacrolimus concentrations and dose seems less evident (see Figure 4). In clinical practice, this means that TDM based on whole blood tacrolimus concentration, especially at higher levels, seems less reliable. In line with this, it was shown that patients with a low hematocrit were likely to have lower whole blood tacrolimus concentrations while the unbound concentration or plasma concentration remained around the population mean [4,41]. This phenomenon could be explained by saturation of the binding receptors for FK506 (tacrolimus) on erythrocytes which are the immunophilins FKBP_12_ and FKBP_13_ [25]. Consequently, in clinical practice, dose adjustment based on whole blood TDM in patients with low hematocrit levels would most likely lead to overdosing and consequently increase the risk of tacrolimus-related toxicity. Another interesting observation is that CYP3A4/5 genetic polymorphisms are associated with variations in the absorption and clearance of tacrolimus during intestinal and hepatic metabolism, resulting in changes to the exposure and starting dose of tacrolimus [42,43]. Since information on these genetic polymorphisms is lacking in the current study cohort, it remains unclear whether these variations are equally detectable in plasma tacrolimus levels. Interestingly, CYP3A5 genetic polymorphisms have so far not been shown to influence IPV of whole blood tacrolimus [44]. Future studies should investigate whether the IPV of plasma tacrolimus concentrations is affected by these genetic polymorphisms.

In the current study, we found, compared to whole-blood tacrolimus levels, a statistically significant association of plasma tacrolimus levels with lower levels of PCS and MCS (HRQoL) in KTRs. Considering the pharmacokinetics of tacrolimus, our findings confirm the important role of the free fraction of tacrolimus as the effective part, which was previously shown to be linearly related to plasma tacrolimus concentrations [4]. Measuring plasma tacrolimus concentrations might more accurately reflect toxicologically relevant concentrations of the drug and may therefore better correlate with clinical outcomes such as HRQoL. Interestingly, several studies have suggested the use of hematocrit-corrected whole blood concentrations in clinical practice, considering the bio-analytical challenges of determining plasma tacrolimus concentrations due to the susceptibility to hemolysis [4,45]. However, as shown in a sensitivity analysis of the current study, plasma tacrolimus concentrations were statistically significantly associated with worse PCS and MCS of HRQoL, whereas hematocrit-corrected whole blood levels did not exhibit any association with HRQoL (see Appendix A, Table A2). Previous studies have shown that many adverse effects of tacrolimus are dose-related, i.e., nephrotoxicity, neurotoxicity, glucose metabolism disturbances, and gastrointestinal disturbances [46]. In line with existing literature, we showed that fatigue and reduced kidney function partially mediated the relationship between plasma tacrolimus concentrations and HRQoL in KTRs. To summarize, our data may suggest the superiority of measuring plasma tacrolimus concentrations for TDM in minimizing post-transplant side effects. However, future research is needed to investigate the relationship between plasma tacrolimus concentrations and the main therapeutic goals of preventing graft rejection and ensuring graft survival.

This study is a first step in investigating whether the above-described non-linear relationship between whole blood and plasma tacrolimus levels is reflected in tacrolimus-related toxicity as expressed by HRQoL. Important to note is that several medications commonly used among KTRs, including antifungal agents and calcium channel blockers, can affect tacrolimus concentrations, which should be considered when interpreting the results [47]. Medication use in this study population is displayed in Appendix D (Table A13).

A limitation of the current study was that the cross-sectional study design did not allow the analysis of causal relationships between tacrolimus levels and quality of life; longitudinal studies are required to assess the effect of the treatment. Tacrolimus concentrations are known to vary widely between and within patients, and a single trough concentration is limited in estimating the area under the blood concentration–time curve (AUC), which reflects total drug exposure. However, trough concentrations are used in common practice and have been correlated with clinical outcomes, where the patients in this cohort are at least 12 months after transplantation and therefore regarded as relatively stable [48,49]. For the subanalysis of IPV, it is important to note that we had a relatively low number of patients with repeated measurements at two time points. For future research, analyzing IPV for plasma tacrolimus concentrations and transplantation outcomes, including HRQoL and rejection, with a larger patient cohort and more frequent plasma tacrolimus measurements post-transplantation would be of great interest. To confirm the clinical significance of the findings, future studies should seek to confirm the potential superiority of plasma tacrolimus concentrations for TDM in a prospective study comparing patient outcomes of groups whose TDM is either based on whole blood or plasma tacrolimus concentrations, where blood is sampled on multiple time points after drug administration.

## 5. Conclusions

Plasma tacrolimus concentrations were statistically significantly associated with worse HRQoL in KTRs, contrary to the current clinical standard of measuring whole blood tacrolimus levels. This suggests that TDM with plasma tacrolimus concentrations may offer an opportunity to reduce the occurrence of tacrolimus-related toxicity.

## Figures and Tables

**Figure 1 pharmaceutics-17-00590-f001:**
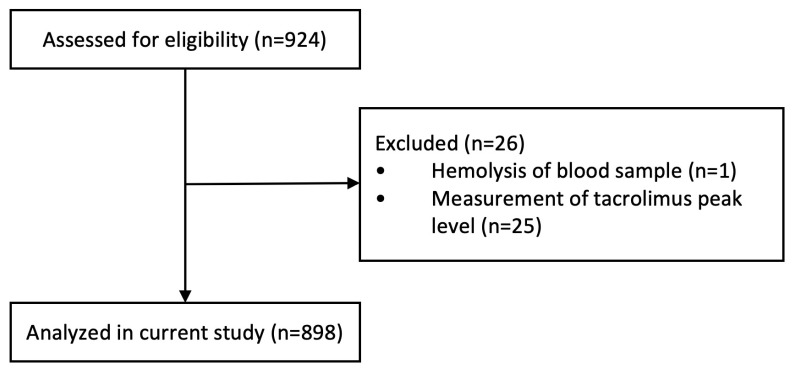
Flowchart.

**Figure 2 pharmaceutics-17-00590-f002:**
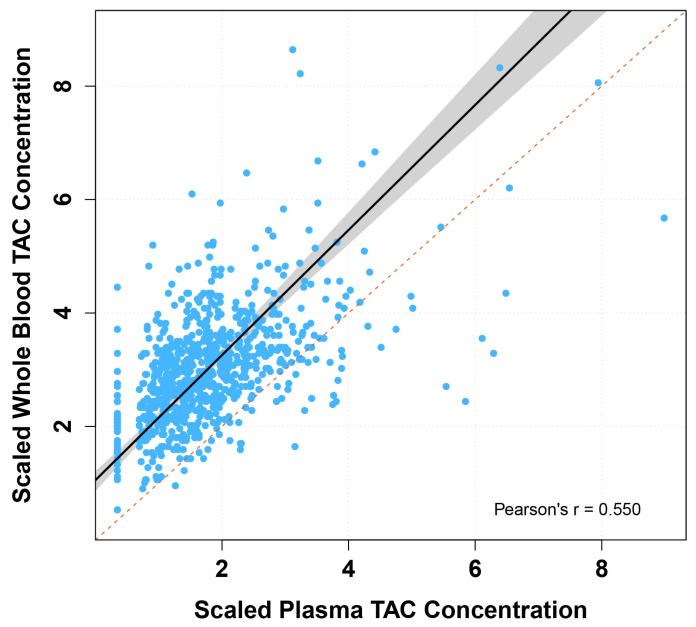
Passing–Bablok regression fit of scaled whole blood versus plasma tacrolimus concentrations. TAC: tacrolimus. Passing–Bablok fit of scaled tacrolimus plasma concentration versus scaled tacrolimus whole blood concentration (solid black line), with 95% confidence interval (grey area) and line of identity (dotted red line).

**Figure 3 pharmaceutics-17-00590-f003:**
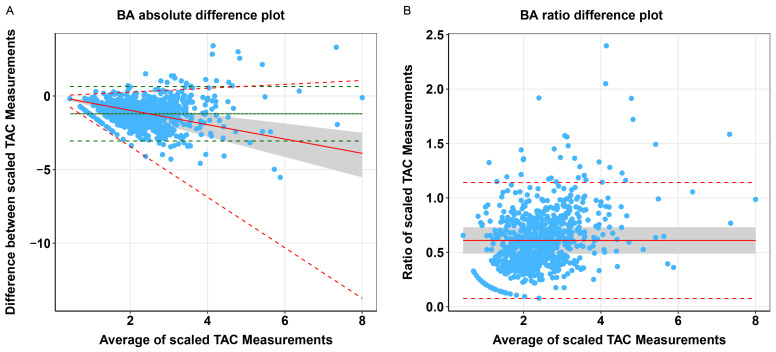
Comparison of scaled whole blood vs. plasma tacrolimus concentrations. BA: Bland–Altman, TAC: tacrolimus. (**A**) Bland–Altman absolute difference plot of scaled tacrolimus whole blood vs. scaled plasma concentrations in KTRs, with mean difference (solid green line) and upper and lower limits of agreement (dashed green lines) based on the standard Bland–Altman formulae, with difference as a linear function of the mean (dotted grey line), predicted difference corrected for heteroscedasticity using the ratio of the data (solid red line), and the corresponding upper and lower limits of agreement (dashed red lines). (**B**) Bland–Altman ratio difference plot of scaled tacrolimus whole blood vs. scaled plasma concentrations in KTRs, with mean difference (solid red line) and upper and lower limits of agreement (dashed red lines). N.B. The grey area represents the ±20% limits around the mean difference between scaled tacrolimus whole blood and plasma concentrations.

**Figure 4 pharmaceutics-17-00590-f004:**
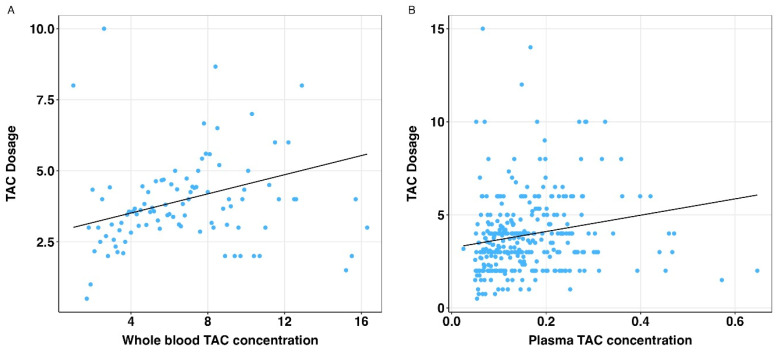
Relation of whole blood and plasma tacrolimus concentrations and daily tacrolimus dose. TAC: tacrolimus. (**A**) Scatterplot with regression line (solid black line) for association between whole blood TAC concentrations and daily TAC dosage. (**B**) Scatterplot with regression line (solid black line) for association between plasma TAC concentrations and daily TAC dosage.

**Figure 5 pharmaceutics-17-00590-f005:**
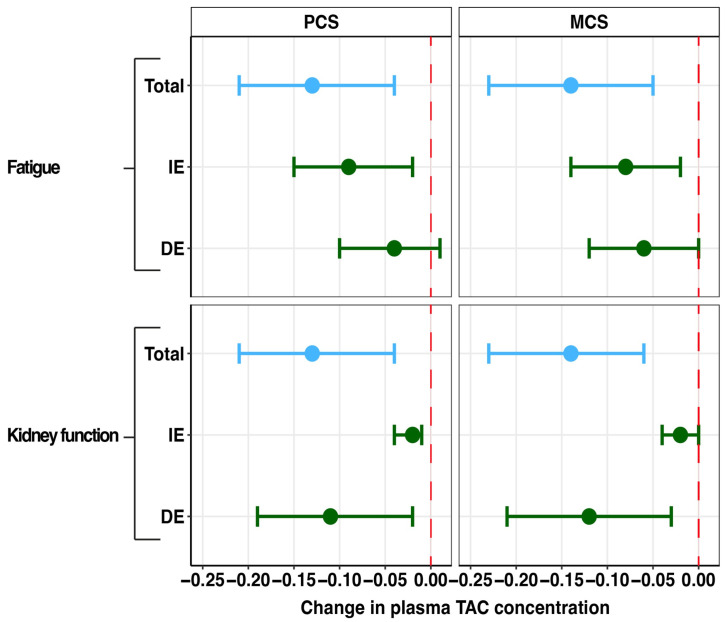
Forest plot of mediation effects of tacrolimus side effects on the associations of plasma tacrolimus with physical and mental component scores. PCS: physical component summary, MCS: mental component summary, DE: direct effect, IE: indirect effect, TAC: tacrolimus. Data are presented as point estimates (linear regression beta coefficients) and 95% confidence intervals (beta coefficient ±1.96 (standard error [beta coefficient]).

**Table 1 pharmaceutics-17-00590-t001:** Baseline characteristics of study population.

**N**	**898**
**Demographics**	
Age, years	55.1 ± 13.8
Sex, n (%) female	333 (37.1)
Time since Tx, years	1.03 [1.00–3.72]
**Tacrolimus**	
Dose TAC, mg/day ^1^	3.0 [2.0–4.0]
Whole blood TAC levels, µg/L ^2^	5.7 ± 1.9
Plasma TAC levels, µg/L ^3^	0.13 ± 0.07
**HRQoL**	
PCS ^4^	47.1 ± 9.5
MCS ^5^	51.3 ± 8.0
**Tacrolimus-related side effects**	
Chronic fatigue, CIS20R ^6^	26.6 ± 13.2
Sleep quality, PSQI ^7^	5.0 [3.0–7.0]
Shannon diversity index ^8^	2.3 ± 0.5
Water percentage stool sample ^9^	75.1 ± 6.6
Kidney function, eGFR	53.5 ± 18.7
PTDM ^10^	171 (19.7)
Alopecia, MTSOSD-59R ^11^	256 (38.4)
Tremor, TRS-C ^12^	0 [0–9]

TAC: tacrolimus, HRQoL: health-related quality of life, PCS: physical component summary, MCS: mental component summary, PTDM: post-transplantation diabetes mellitus. Missing data (N): ^1^ 31; ^2^ 35; ^3^ 92; ^4^ 254; ^5^ 254; ^6^ 244; ^7^ 282; ^8^ 399; ^9^ 598; ^10^ 31; ^11^ 231; ^12^ 657.

**Table 2 pharmaceutics-17-00590-t002:** Comparison between scaled whole blood and plasma tacrolimus concentrations.

**Passing–Bablok Regression Analyses**					
	**N**	**Intercept (95% CI)**		**Slope (95% CI)**	
KTR	797	1.05 (0.89–1.22)		1.10 (1.00–1.21)	
**Bland–Altman Analyses**					
**Bland–Altman Absolute Differences**			**Bland–Altman Ratio Differences**		
**Bias (95% CI)**	**95% LOA (bias ± 1.96 SD)**	** *p* ** **-Value**	**Bias (95% CI)**	**95% LOA (bias ± 1.96 SD)**	** *p* ** **-Value**
0.002 (−0.07–0.08)	−3.06–0.64	0.96	0.12 (0.10–0.14)	0.08–1.14	<0.001

TAC: tacrolimus, CI: confidence interval, LOA: limits of agreement, SD: standard deviation.

**Table 3 pharmaceutics-17-00590-t003:** Association of whole blood and plasma TAC concentrations with daily tacrolimus dose.

Whole Blood TAC				Plasma TAC				*p*-Value ^1^
N	Coef β (95% CI)	St. β	*p*-Value	N	Coef β (95% CI)	St. β	*p*-Value	
858	0.17 (0.09–0.26)	0.14	<0.001	803	4.43 (2.17–6.70)	0.13	<0.001	0.54

TAC: tacrolimus. Outcome variable: daily TAC dose. Linear regression analyses, normal distribution of residuals was checked. ^1^ Testing proportional difference of bootstrap standardized β (1000 replications).

**Table 4 pharmaceutics-17-00590-t004:** Association of whole blood and plasma TAC concentrations with HRQoL.

	Whole Blood TAC				Plasma TAC				*p*-Value ^1^
	N	Coef β (95% CI)	St. β	*p*-Value	N	Coef β (95% CI)	St. β	*p*-Value	
PCS	618	−0.14 (−0.55–0.28)	−0.03	0.52	593	−16.94 (−27.97–5.90)	−0.12	0.003	0.01
MCS	618	0.004 (−0.35–0.36)	0.001	0.98	593	−16.01 (−25.34–6.68)	−0.14	<0.001	<0.001

PCS: physical component summary, MCS: mental component summary, TAC: tacrolimus. Outcome variable: PCS or MCS, HRQoL. Linear regression analyses, normal distribution of residuals was checked. ^1^ Testing proportional difference of bootstrap standardized β (1000 replications).

## Data Availability

The datasets generated during and/or analyzed during the current study are not publicly available due to the sensitivity of the data and the restrictions from the informed consent but are available from the corresponding author on reasonable request.

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
