# Peer review of "Plasma Versus Whole Blood Tacrolimus Concentrations and Health-Related Quality of Life in Kidney Transplant Recipients"

_pharmaceutics, 2025, doi:10.3390/pharmaceutics17050590_

Round 1
Reviewer 1 Report
Comments and Suggestions for Authors
Dear authors,
this is interesting topic since the importance of tacrolimus for solid organ transplantation.
I have a few comments:
- Can you describe the complete protocol with dosing regimens and what kind of Tac formulation were prescribed?
- What other drugs (beside of immunosuppresants) were used by patients? Is there potential interactions?
- Did you use one or more concentrations per patient? Is there chance to calculate intrapatient variability from plasma concentrations?
- We all know about the association between CYP3A5* and CYP3A4*22 SNPs and whole blood concentration. Can you tell what about concentration in plasma or C0/D derived from plasma? Even hypothetical.
- How can you explain the significant association between HRQOL and plasma, but not whole blood concentration of Tac. More detailed?
Author Response
Comment 1: Can you describe the complete protocol with dosing regimens and what kind of Tac formulation were prescribed?
Response 1: We thank the reviewer for the comment. At the University Medical Center Groningen, tacrolimus therapy after kidney transplantation starts on the day of operation. Since 2022, the first choice has been slow-release tacrolimus (Envarsus or Advagraf), and the second choice has been short-acting tacrolimus (Prograf). This means that for the largest part of this cohort, the first choice was the Prograf formulation, which is reflected in the numbers diplayed below.
If chosen for Envarsus, the following protocol was handled:
- Day 0: 7 mg pre-operative
- Day 1-4: 7 mg once daily (at least 12 hours after the first administration)
- From day 5 onwards: target levels of 8-12 ug/mL
- From week 7 onwards: target levels of 6-10 ug/mL
- From 6 months onwards: target levels of 4-6 ug/mL
If chosen for Advagraf, the following protocol was handled:
- Day 0: 10 mg pre-operative
- Day 1-4: 10 mg once daily (at least 12 hours after the first administration)
- From day 5 onwards: target levels of 8-12 ug/mL
- From week 7 onwards: target levels of 6-10 ug/mL
- From 6 months onwards: target levels of 4-6 ug/mL
If chosen for Prograf, the following protocol was handled:
- Day 0: 0.075 mg/kg pre-operative
- Day 1-4: 0.075 mg/kg twice daily
- From day 5 onwards: target levels of 8-12 ug/mL
- From week 7 onwards: target levels of 6-10 ug/mL
- From 6 months onwards: target levels of 4-6 ug/mL
In the current study, KTR were included at least 12 months after transplantation; thus, all included KTR were on a regimen with a target trough level of 4-6 ug/mL, as described in the Methods section (Lines 105-107).
The distribution of tacrolimus formulation prescribed in this cohort is displayed in the following Table:
Tacrolimus formulation |
N (%) |
Envarsus |
89 (9.9) |
Advagraf |
224 (24.9) |
Prograf |
573 (63.8) |
Unknown |
12 (1.3) |
The distribution of prescribed Tacrolimus formulation was added to the Results section (Lines 272-274).
Comment 2: What other drugs (beside of immunosuppressants) were used by patients? Is there potential interactions?
Response 2: We thank the reviewer for this question. The table below provides a detailed overview of medication use in the study population. Among the medications used, there are potential agents that could interact with tacrolimus, specifically CYP3A and/or P-glycoprotein inhibitors or inducers such as antifungal agents [1]. This interaction could affect the bioavailability or clearance of tacrolimus or both, which would theoretically affect whole blood and plasma tacrolimus concentrations likewise. We added the table showing medication use to the Appendix and included the following part in the discussion: Important to note is that several medications commonly used among KTR, including antifungal agents and calcium channel blockers, can affect tacrolimus concentrations, which should be taken into account when interpreting the results. Medication use in this study population is displayed in Appendix D (Table D1) (Lines 444-448).
Reference:
- Kirubakaran R, Stocker SL, Hennig S, Day RO, Carland JE. Population Pharmacokinetic Models of Tacrolimus in Adult Transplant Recipients: A Systematic Review.Clin Pharmacokinet. 2020;59(11):1357-1392. doi:10.1007/s40262-020-00922-x
Other immunosuppressants |
N (%) |
Prednisolone |
873 (97.3) |
Mycophenolate mofetil |
718 (80.0) |
Azathioprine |
51 (5.7) |
MTOR inhibitors (everolimus or sirolimus) |
42 (4.7) |
Antihypertensives |
|
Beta-blockers |
538 (60.0) |
Calcium channel blockers |
407 (45.4) |
Alpha blockers |
136 (15.2) |
Diuretics |
213 (23.8) |
Angiotensin-converting enzyme inhibitors |
173 (19.3) |
Angiotensin II receptor blockers |
111 (12.4) |
Lipid-lowering treatment |
|
Statins |
508 (56.6) |
Gemfibrozil |
50 (5.6) |
Ezetimibe |
16 (1.8) |
Antidiabetic Medication |
|
Metformin |
100 (11.2) |
Sulfonylureas |
53 (5.9) |
SGLT2 inhibitors |
3 (0.3) |
DPP4 inhibitors |
10 (1.1) |
GLP-1 receptor agonists |
5 (0.6) |
Insulin |
121 (13.5) |
Anticoagulants |
|
Acetylsalicylic acid |
188 (21.0) |
Dipyridamole |
5 (0.6) |
Clopidogrel |
48 (5.4) |
Apixaban |
12 (1.3) |
Non-vitamin K oral anticoagulants |
12 (2.0) |
Vitamin K antagonists |
105 (11.7) |
Gastroprotective agents |
|
Proton pump inhibitors |
661 (73.7) |
H2-receptor antagonists |
32 (3.6) |
Antacids |
11 (1.2) |
Mucosal protective agents |
3 (0.3) |
Antibiotics |
|
Sulfonamides |
35 (3.9) |
Fluoroquinolones |
14 (1.6) |
Macrolides |
10 (1.1) |
Penicillin |
2 (0.2) |
Antifungal agents |
|
Imidazoles |
16 (1.8) |
Antidepressant Medication |
|
Tricyclic antidepressants |
22 (2.5) |
Selective serotonin reuptake inhibitors |
29 (3.2) |
Sleep medication |
|
Benzodiazepines |
65 (7.3) |
Melatonin agonists |
3 (0.3) |
Comment 3: Did you use one or more concentrations per patient? Is there chance to calculate intrapatient variability from plasma concentrations?
Response 3: We thank the reviewer for this question. In the first version of the manuscript, we did not include multiple measurements per patient. We, however, agree with the reviewer that analysing the intra-patient variability (IPV) for tacrolimus plasma concentrations is informative. In tacrolimus whole blood measurements, higher IPV is associated with worse outcomes, such as decreased graft survival [1,2]. In the TransplantLines cohort, two repeated tacrolimus plasma concentration measurements were available for 162 patients. Unfortunately, no repeated measures of Health-Related Quality of Life were available. These patients provided blood samples at 12 and 24 months after transplantation. There was no significant difference in the plasma concentration of tacrolimus at 12 and 24 months after transplantation (Wilcoxon signed-rank test, V=8734, P=0.20; Figure B1-A). IPV was calculated by dividing the standard deviation by the mean of both measurements for each patient and multiplying by 100. This provided a coefficient of variability (COV) per patient. A total of 71 patients had a COV > 30%, which is considered a high IPV (Figure B1-B) [3]. Next, we analyzed if COV was associated with HRQoL by correlating COV with PCS and MCS. COV was neither significantly associated with the PCS nor with the MCS (r=0.01, P=0.88 and r=0.04, P=0.59, respectively, Figure B1-C and D). To accommodate the comment of the reviewer, we added this subanalysis to Lines 343-351 of the Results section and to the Methodology section (Lines 234-239). It should be noted that we had a relatively low number of patients with repeated measurements at two time points. It would be interesting to analyze IPV for plasma tacrolimus concentrations and transplantation outcomes such as HRQoL and rejection with a higher number of patients and plasma tacrolimus measurements at a higher frequency post-transplantation, which was mentioned in Lines 457-461 of the Discussion section.
Figure B1. Overview of the intra-patient variability for plasma tacrolimus. (A) Boxplot depicting plasma concentrations of tacrolimus for repeated measurements of 162 patients. (B) Histogram depicting the coefficient of variability for repeated plasma tacrolimus measurements. (C and D) Correlation plot for the coefficient of variability and the PCS and MCS.
PCS: physical component summary, MCS: mental component summary, TAC: tacrolimus, COV: coefficient of variability.
References:
- Rozen-Zvi B, Schneider S, Lichtenberg S, et al. Association of the combination of time-weighted variability of tacrolimus blood level and exposure to low drug levels with graft survival after kidney transplantation.Nephrol Dial Transplant. 2017;32(2):393-399. doi:10.1093/ndt/gfw394
- Coste G, Lemaitre F. The Role of Intra-Patient Variability of Tacrolimus Drug Concentrations in Solid Organ Transplantation: A Focus on Liver, Heart, Lung and Pancreas.Pharmaceutics. 2022;14(2):379. Published 2022 Feb 8. doi:10.3390/pharmaceutics14020379
- Piburn KH, Sigurjonsdottir VK, Indridason OS, et al. Patterns in Tacrolimus Variability and Association withDe Novo Donor-Specific Antibody Formation in Pediatric Kidney Transplant Recipients. Clin J Am Soc Nephrol. 2022;17(8):1194-1203. doi:10.2215/CJN.16421221
Comment 4: We all know about the association between CYP3A5* and CYP3A4*22 SNPs and whole blood concentration. Can you tell what about concentration in plasma or C0/D derived from plasma? Even hypothetical.
Response 4: We thank the reviewer for this question. It has been shown that the exposure and starting dose of tacrolimus are significantly influenced by the gene encoding the tacrolimus-metabolizing enzyme cytochromes P450 3A4/5 (CYP3A4/5) [1].
Tacrolimus is primarily metabolised and degraded by the cytochrome P450 system, particularly the CYP3A subfamily. The isoforms CYP3A4 and CYP3A5 are essential for the metabolism of tacrolimus in both the liver and intestines. Individuals carrying a variant in the CYP3A5 gene, specifically the CYP3A5*3 allele, exhibit an RNA splicing defect that results in inadequate enzyme expression. As a result, individuals with no enzyme expression (CYP3A5*3/*3) demonstrate increased bioavailability and exposure to tacrolimus due to enhanced intestinal absorption. Conversely, individuals possessing at least one functional CYP3A5*1 allele produce CYP3A5 and therefore require higher doses of tacrolimus to reach target concentrations. [2].
Tacrolimus is metabolized in the liver by CYP3A4 with significant inter-individual variation in the activity of this enzyme. The functional polymorphism of the CYP3A4*22 allele is correlated with a reduced expression level of CYP3A4 mRNA and protein. This relationship results in a decline in the overall activity of the enzyme, which may influence the clearance of tacrolimus [2].
The relationship of these CYP3A4/5 genetic polymorphisms and tacrolimus absorption and clearance is indeed well-studied for whole blood tacrolimus; however, research on the effects on plasma tacrolimus concentrations is lacking. Unfortunately, information on the presence of the genotypes in KTR included in this study is lacking.
In our study, we demonstrated that high whole blood tacrolimus levels do not necessarily reflect high plasma tacrolimus levels. As outlined in the Discussion section, this is in line with Sikma et al. who showed a non-linear relationship between whole blood and unbound tacrolimus plasma concentrations in recipients after thoracic organ transplantation, especially for higher whole blood concentrations. Therefore, while it seems probable that CYP3A4/5 genetic polymorphisms may affect plasma tacrolimus concentration levels similarly to whole blood levels, as these genotypes primarily influence the absorption and clearance of tacrolimus, it remains unclear how the distribution of unbound/bound tacrolimus might be impacted.
Interestingly, CYP3A5 polymorphisms have so far not been shown to influence intra-patient variability (IPV) of whole blood tacrolimus concentrations [3]. However, it would be of great interest to assess whether IPV of plasma tacrolimus concentrations is influenced by these genetic polymorphisms, since plasma tacrolimus concentrations seem to better represent the active fraction of tacrolimus. This hypothesis warrants further exploration in future research.
To accommodate the reviewer's comment, we included a mention of this hypothesis in the Discussion section (Lines 411-419) of the revised version of the manuscript.
References:
- Mohammed Ali Z, Meertens M, Fernández B, et al.CYP3A5*3 and CYP3A4*22 Cluster Polymorphism Effects on LCP-Tac Tacrolimus Exposure: Population Pharmacokinetic Approach. Pharmaceutics. 2023;15(12):2699. doi:10.3390/pharmaceutics15122699
- Pallet N, Jannot AS, El Bahri M, et al. Kidney transplant recipients carrying the CYP3A4*22 allelic variant have reduced tacrolimus clearance and often reach supratherapeutic tacrolimus concentrations.Am J Transplant. 2015;15(3):800-805. doi:10.1111/ajt.13059
- Nuchjumroon A, Vadcharavivad S, Singhan W, et al. Comparison of Tacrolimus Intra-Patient Variability during 6-12 Months after Kidney Transplantation between CYP3A5 Expressers and Nonexpressers.J Clin Med. 2022;11(21):6320. doi:10.3390/jcm11216320
Comment 5: How can you explain the significant association between HRQOL and plasma, but not whole blood concentration of Tac. More detailed?
Response 5: We thank the reviewer for the question. In our study, we demonstrated that higher plasma tacrolimus concentrations were significantly associated with lower physical and mental component summaries of Health-Related Quality of Life (HRQoL). In contrast, whole blood tacrolimus concentrations did not show such associations. As outlined in the Introduction section, tacrolimus is extensively bound to erythrocytes and circulating plasma proteins with only a small proportion (0.3-2%) circulating as unbound in plasma, known as the free fraction. This unbound fraction is the pharmacologically active component responsible for both therapeutic effects and toxicity. In contrast, whole blood concentrations reflect both bound and unbound tacrolimus, and might therefore not accurately reflect the free active concentration at the site of action. This is supported by our observation that plasma and whole blood tacrolimus levels showed poor agreement. This distinction can be made by measuring plasma tacrolimus concentrations (which as outlined in the Discussion have been shown to be linearly related to the free active fraction), and, therefore, correlating more accurately with clinical outcomes such as HRQoL. Thus, our findings suggest that plasma tacrolimus levels may better reflect toxicologically relevant concentrations of the drug compared to those in whole blood. For example, a patient within the target range of whole blood tacrolimus levels could experience a variety of side effects and consequently a reduced HRQoL. However, in clinical practice, their dose might remain unchanged to avoid subtherapeutic levels, based on opinions formed from therapeutic drug monitoring of whole blood tacrolimus levels. Future research, particularly longitudinal studies, should explore the causal pathways between tacrolimus exposure, side effects, and HRQoL to better inform individualized dosing strategies. We further elaborated on this explication in the Discussion section (Lines 424-427).
Reviewer 2 Report
Comments and Suggestions for Authors
Plasma versus whole blood tacrolimus concentrations and health-related quality of life in kidney transplant recipients.
Manuscript ID: pharmaceutics-3577920
The current study investigated whether plasma tacrolimus levels correlated more strongly with HRQoL than whole blood tacrolimus levels in KTR. The authors compared plasma and whole blood tacrolimus levels and daily tacrolimus doses. Then they compared the relationship between plasma and whole blood tacrolimus levels and HRQoL and explored the mediating effect of various tacrolimus-related side effects.
Through an extensive study design and analytical methods as compared with whole blood tacrolimus concentrations, plasma tacrolimus concentrations exhibited a negative association with HRQoL in KTR. Consequently, therapeutic drug monitoring using plasma tacrolimus concentrations may reduce the occurrence of tacrolimus-related toxicity in kidney transplant patients.
Overall, the study is well-designed, and presented in the current manuscript. The introduction, materials and methods section and the results sections are very well explained. Various analytical and statistical methods of PKPD have also been utilized and executed. Overall the manuscript can be published in its current version.
Minor comments:
- Ensure that all the references are recent within a 5-10 year range.
Author Response
Comment 1: Ensure that all the references are recent within a 5-10 year range.
Response 1: We thank the reviewer for this comment. If the references used did not address methodological issues, we substituted them with more recent publications from the past 10 years (see References).
Reviewer 3 Report
Comments and Suggestions for Authors
The present study aimed to investigate the association of plasma tacrolimus levels (in comparison to whole blood tacrolimus concentrations) with health-related quality of life (HRQoL) in kidney transplant recipients. It is a very well-designed study with a significant number of subjects included. The methods are appropriate for this type of study, including statistical analyses. The results obtained are novel and significant, and they have been adequately discussed in the manuscript.
In my opinion, the manuscript could be accepted in the current form; however, I have only a few suggestions.
The addition of titled subsections in the Methods section would improve the organization of the manuscript and make it easier to follow.
Line 97: “if peak levels of tacrolimus were determined” – please, clarify.
Please clarify in the text why the number of KTRs in Table 2 is 797. Also, the number of subjects stated in tables 3 and 4 should be clarified.
Author Response
Comment 1: The addition of titled subsections in the Methods section would improve the organization of the manuscript and make it easier to follow.
Response 1: We thank the reviewer for this suggestion. In response, we have added titles to the subsections of the Methods section to improve its structure and readability.
Comment 2: Line 97: “if peak levels of tacrolimus were determined” – please, clarify.
Response 2: We thank the reviewer for this comment. As described in Lines 111-116, participants were instructed to administer tacrolimus at 10:00 am for a once-daily dosing regimen and at 10:00 am and 10:00 pm for a twice-daily dosing regimen. Blood samples were collected at 8:00 am to obtain tacrolimus trough concentrations, which aligns with standard clinical practice for therapeutic drug monitoring. If it was noted during measurement, or indicated by the participant, that tacrolimus had been administered shortly before blood sampling and thus resulted in a peak level, these samples were excluded from the analyses to ensure consistency of tacrolimus concentrations.
Comment 3: Please clarify in the text why the number of KTRs in Table 2 is 797. Also, the number of subjects stated in tables 3 and 4 should be clarified.
Response 3: We thank the reviewer for this comment. As noted in the description of Table 1, some data were missing due to logistical constraints inherent in conducting a large cohort study. Among the 898 included kidney transplant recipients, we are missing 35 whole blood tacrolimus samples, 92 plasma tacrolimus samples, 31 data points on tacrolimus dose at the time of inclusion, and 254 responses to the SF-36 questionnaire, which translates to missing data points in the physical and mental component summaries. However, the missing data points across different variables are not always absent for the same subject. Therefore, we chose not to exclude participants with partial data in order to maintain statistical power for the various analyses.
This translates to the different numbers of KTR for the different analyses as indicated in the Tables.
Table 2 shows the comparison between whole blood and plasma tacrolimus concentrations. Therefore, for this analysis, only KTR without missing data in these variables were included, resulting in a total of 797 KTR. The same is true for Table 3, which shows the association of whole blood and plasma tacrolimus concentrations with the daily tacrolimus dose, and Table 4, which shows the association of whole blood and plasma tacrolimus with Health-related Quality of Life (the variables physical component summary and mental component summary).
To clarify this in the manuscript, we added the following sentence to the Method section: Due to logistical constraints inherent in conducting a large cohort study, some data points were missing for certain KTR (as outlined in Table 1). For each analysis, only participants with complete data relevant to that specific analysis were included, resulting in varying sample sizes reported for each analysis. (Lines 253-257)